# What Makes Hotel Chefs in Korea Interact with SNS Community at Work? Modeling the Interplay between Social Capital and Job Satisfaction by the Level of Customer Orientation

**DOI:** 10.3390/ijerph17197129

**Published:** 2020-09-29

**Authors:** Sang-Won Seo, Hyeon-Cheol Kim, Zong-Yi Zhu, Jung-Tak Lee

**Affiliations:** 1Graduate School of Tourism & Hospitality, Kyonggi University, Seoul 03746, Korea; wons_99@naver.com; 2School of Business Administration, College of Business and Economics, Chung-Ang University, Seoul 06974, Korea; hckim@cau.ac.kr; 3Department of Arts and Cultural Management, Graduate School, Chung-Ang University, Seoul 06974, Korea; vampirenylon@cau.ac.kr; 4Department of Culinary Arts, Ansan University, Ansan 15328, Korea

**Keywords:** hotel chefs, social capital, social presence, job satisfaction, customer orientation

## Abstract

This study aimed to investigate the effect of social network services (SNS) on hotel chef job satisfaction, and to provide an effective strategy to reduce chef turnover and maintain sustainable economic development in hospitality. The intention was to empirically test and analyze the effect of SNS on hotel chef job satisfaction by applying the social capital theory. The social capital theory was explored and the effect of chefs’ social capital on their social presence and job satisfaction was demonstrated. Furthermore, this study aimed to determine the moderation effect of customer orientation. A total of 130 surveys were collected from chefs working at Michelin-starred restaurants in Seoul, Korea. SPSS and AMOS were used to conduct statistical analyses. The outputs included exploratory factor analysis, confirmatory factor analysis, convergent analysis, discriminant analysis, path analysis, mediation effect analysis, and moderation effect analysis. The results illustrated that bridging social capital significantly impacts chef social presence, while bonding social capital does not significantly influence their presence. In addition, both bonding and bridging social capital positively relate to chef job satisfaction. Significant mediation and moderation effects were demonstrated on the path taken by chefs. The results of this study offer theoretical and managerial implications for hotel human resources managers to enhance chef job satisfaction.

## 1. Introduction

The hospitality and tourism industry is economically significant for many areas of South Korea. In particular, the food and beverage industry provides positive economic effects [1]. For sustainable economic development, it is critical to investigate the factors that affect chef job satisfaction to reduce chef turnover. According to previous studies, chef job satisfaction is impacted by intrinsic and extrinsic factors. Intrinsic factors include job performance and career experience [1,2], while extrinsic factors consist of working conditions, organization commitment, and colleague relationships, which positively influence chef job satisfaction [3,4,5]. Because chefs are critical to the success of food businesses, enabling them to find satisfaction in their professional competences and in their job is important for hospitality industry managers.

However, several studies have suggested that chefs often work in stressful, highly bureaucratic, unsociable, unstable, and unpleasant work environments [6,7]. Previous research has argued that, as customer orders stream out of the printer on the pass table, the head chef calls them out to the sous chefs who, together with commis chefs, immediately begin preparing the meals [8]. If a disharmonious relationship emerges in the chef’s team, it will negatively impact their job performance and job satisfaction. In this regard, a chef’s relationships with their colleagues, supervisor, and organization have a critical effect on chef job performance and job satisfaction. Previous studies have also shown that chefs are less satisfied with their job due to the low professional career development potential in the hospitality industry [9]. In addition, Barnett and Bradely (2007) argued that a good means of retaining qualified employees is by helping them to exchange food knowledge to develop their own careers in the hospitality industry [10]. Thus, social relationships and knowledge exchange are critical factors in maintaining job satisfaction.

With the wide usage of social network services (SNS), consumers can easily access their online social connections at anytime and anywhere using their smart devices. At the beginning of the third decade of the twenty-first century, SNSs are, increasingly, becoming important platforms for the creation and maintenance of social capital [11,12]. It has also been determined that SNS community participants positively increase their social capital [13]. The chef SNS community is also regarded as an important way to develop relationships with other chefs and facilitate recipe knowledge exchange [14]. Following the social capital theory, in the organization literature it has been demonstrated that social capital in the organization positively impacts the job performance and job satisfaction of the employee [15,16,17]. Following the previous studies discussed above, social capital is a critical factor in influencing employee social presence and job satisfaction. Chef SNS community social capital could assist chefs to communicate with others and positively impact their social presence and job satisfaction.

However, most chef job satisfaction studies have focused on the relationship between chef job stress and job satisfaction [18]. Previous research has also investigated the effect of job crafting on chef job satisfaction [19]. In addition, previous studies have determined hotel chef job’s motivational factors and the effect on their job satisfaction [20]. Fewer studies have been undertaken on the relationship between chef social capital, social presence, and job satisfaction. This study intends to fill this gap and determine the effect of social capital on social presence and job satisfaction.

Moreover, chefs are service workers. They have to enhance their experience to satisfy their needs [21]. Hotel chefs can be divided into those with high consumer orientation and those with low consumer orientation. High consumer orientation chefs try to improve their cooking skill to offer consumers a more satisfying experience. To improve their skills, high consumer orientation chefs are active participants in SNS community activity to develop their social capital with other chefs. Previous studies have demonstrated that service employee consumer orientation has different impacts on their job satisfaction [22]. However, these studies have paid little attention to investigate the difference in customer orientation moderation effect.

Based on previous studies, little research has examined the effect of social capital on chef job satisfaction. In addition, although service workers are required to have high customer orientation, previous studies paid little attention to the effect of customer orientation. The purpose of this study is to fill the gap of previous studies. This study intends to identify the effect of chef SNS social capital on their social presence and job satisfaction. This study collected data from S-Hotel in Seoul, Korea to investigate the research question. The specific research aims were as follows: First, to determine the effect of social capital on social presence and job satisfaction; second, to investigate the moderation effect of customer orientation; and third, to demonstrate the social presence mediation effect between social capital and job satisfaction. The results of this study will enhance our understanding of chef job satisfaction by clarifying social capital. This study could offer guidelines for human resources management of hotels to improve chef job satisfaction. In addition, these results could provide an effective strategy to reduce chef turnover and to maintain sustainable economic development.

## 2. Theoretical Background and Hypotheses Development

### 2.1. Factors Affecting Chef Job Satisfaction

Job satisfaction refers to a worker’s feelings about their job [23]. Thus, job satisfaction could be explained as the perceived relationship between the expectation of an employee regarding their job and what they actually obtain from that job, in addition to the importance or value that an employee attributes to their work [1]. According to the organizational behavior literature, job satisfaction is affected by intrinsic and extrinsic factors. Intrinsic factors include job achievement, job independence, and job variety, while extrinsic factors consist of colleague relationships, working conditions, and payment [24]. Previous studies have determined that job performance and career experience are critical intrinsic factors that positively relate to job satisfaction [1,2]. In organization management, extrinsic factors are more important than intrinsic factors. Prior studies found a positive correlation among working conditions, organization commitment, colleague relationships, and job satisfaction. Hospitality research on chef job satisfaction also found the same results. The colleague relationship has been investigated as one of the critical factors affecting chef job satisfaction. Rayton’s (2006) research investigated the perceived relationship with supervisors and found their support positively impacts chef job satisfaction [3]. In addition, boutique hotel chef research determined that there was a positive effect of leadership behavior on job satisfaction [4]. Previous studies also found that support from colleagues and training opportunities are significantly associated with job satisfaction [5]. In this regard, it could be argued social capital is one of the critical factors affecting chef job satisfaction.

### 2.2. Chef SNS Community Participation

The social network service community is widely used for knowledge sharing, social interaction, and developing social relationships. Following previous studies, the SNS community is utilized for knowledge sharing, which could be stored, utilized, and reused for current and future situations [25]. In the service sectors, the SNS community is used for offering available information, which can help others to solve problems and develop new ideas [26]. Information sharing in the SNS community has been predicted as an antecedent of sustainable competitive advantage to raise service worker job performance and company revenues [27,28]. Organization management research has argued that positive SNS community communication behavior can foster relationships with other co-workers and improve their job performance, resulting in a positive impact on company revenue [29,30].

The chef SNS community is an important resource for information and knowledge exchange. In the context of haute cuisine chefs, the domain represents the commonly understood practices of cooking, but also diverse cuisines and new cooking technologies that have been developed [14]. Through social interaction among individuals, they exchange cooking knowledge. The chef SNS community comprises individual chefs who offer cooking information to others and exchange their ideas, which helps the development of social capital with others. A previous study determined that chefs wish to share their recipes with colleagues who they believe adhere to implicit social norms that have been established in the chef community [31]. As a result, the SNS community is an important environment for chefs to build relationships with other chefs. This helps them develop social capital and online behavior.

### 2.3. Social Capital and Social Presence

Social capital means the sum of the resources, actual or virtual, which accrue to an individual or a group by virtue of possessing a durable network of more or less institutionalized relationships of mutual acquaintance and recognition [12]. Previous research has argued that social capital is a relationship that people use as means of fulfilling their needs. Moreover, social capital also entails interpersonal relationships in organizations for receiving benefits [32]. Thus, social capital is a critical factor for individuals to improve their social presence and job satisfaction.

SNSs are not only used for connecting offline friends and colleagues, but also for expanding online relationships through virtual social interaction with others who share the same or similar interests [12]. Positive social capital can increase satisfaction and civic participation [33]. Social capital consists of bridging social capital and bonding social capital [34]. Bridging social capital and bonding social capital are opposing perspectives. Bridging social capital refers to weak relationships, which offer available opportunities for information sharing. On the contrary, bonding social capital refers to strong relationships, which provide social support and trust [12]. A recent study determined that the SNS is a useful medium for achieving bridging social capital and bonding social capital. Previous research determined that relationship maintenance behaviors on Facebook result in an increase in social capital [33]. Human resource research also indicated that the percentage of coworker connections within the total number of Facebook contacts has a positive influence on coworker social capital [34]. Thus, online communication can help users to enhance their social capital.

Social presence is a critical outcome of social capital. Social presence refers to a consumer’s perception regarding the degree of service they received. According to a previous study, social presence consists of two criteria. The first refers to the sense of strong relationship and the second is the psychological distance of communication. In the research of Ozatok Murat et al. (2015), social capital theory was employed to understand the social presence of students in online learning environments and illustrated that social presence related more to communication between weak ties rather than within strong-tied subsets of participants [35]. As Shin (2013) explained, SNS users utilized social media to have social interactions with other users or online content producers, which helps users to build social capital with others and then facilitate their social presences [36]. However, social capital consists of bridging social capital and bonding social capital, which are opposing perspectives. Bridging social capital refers to weak relationships. On the contrary, bonding social capital refers to strong relationships. In addition, it was recently found that bonding social capital refers to strong relationships, which may impact their offline communication and reduce their online social presence. However, because bridging social capital involves weak relationships, online communication may enhance their relationship, which may positively impact social presence [37]. In this perspective, this study argued that chefs’ SNS usage would help them develop social capital and, as a result, impact their social presence. Thus, the hypotheses could be stated as follows:

**Hypothesis** **1 (H1).**
*Social capital bonding is positively associated with social presence.*


**Hypothesis** **2 (H2).**
*Social capital bridging is positively associated with social presence.*


### 2.4. Social Presence and Job Satisfaction

Social presence is a concept which is based in telecommunications research. Social presence theory is used for analyzing the social psychological dimensions of mediated communication [38]. Social presence theory argues that the quality of the medium itself and the ability to assert varies in the degree of social presence [39]. According to previous studies, social presence consists of two dimensions. The first dimension describes the sense of close connection one feels in a relationship [40]; the other one describes the psychological distance between a communicator and the recipient of the communication [41]. Other researchers describe the qualities of media technology that may have an effect on the degree of presence. In this case, social presence means individuals perceived by others as “real” while they are in the context of their mediated communication [39]. According to this perspective, social presence is used to describe the sense of relationship and perception of others as “real”. With the development of information technology, a consumer can easily have interaction or communication with others in the online environment.

Recently, online messaging was developed for business and has been widely used across industries. According to a previous study, the aspect of employee social presence consists of three dimensions, which are relationship, psychological, and interaction effect [42,43,44]. Previous research indicated that social presence positively impacts employee job satisfaction [45]. To improve business efficiency, strong relationships and perceiving others as “real” appear more important than face-to-face communication. In particular, in the food service industry, in contrast to serving which involves interaction between consumers and other employees, chefs work in a separated space with other employees. In this way, online interaction appears to be more important for them. Developing strong relationships and perceiving others as real may be critical factors to improving their job satisfaction.

Several studies have determined the positive relationship between social presence and satisfaction. The Ellison et al. (2007) study found that building relationships with others positively influences social presence, which enhances job satisfaction [46]. Reinout et al. (2006) and Matler and Renzl (2006) determined that social presence is an outcome of interaction assessment which positively impacts co-worker relationships [45,47]. Thus, this study argues that social presence positively influences job satisfaction. The hypothesis is as follows:

**Hypothesis** **3 (H3).**
*Social presence is positively associated with job satisfaction.*


### 2.5. Social Capital and Job Satisfaction

Job satisfaction is a pleasurable or positive emotional state resulting from the appraisal of one’s job or job experience [48]. According to previous research, job satisfaction is positively affected by three dimensions: The employee–organization relationship, the employee–supervisor relationship, and the employee–coworker relationship [49]. The employee–organization relationship helps the employee realize the importance of employee identification with and commitment to organizational strategy and company goals [50]. The employee–supervisor relationship focuses on supervisor delegates and gives autonomy to employees to greatly influence employees’ assessments of their jobs [51]. Lastly, the employee–coworker relationship is also an important factor which can influence job satisfaction. Thus, we can argue that the SNS could help coworkers enhance their social capital, resulting in positive outcomes, such as job performance and job satisfaction. In this perspective, relationships within an organization are a critical factor influencing employee job satisfaction.

With the wide usage of SNSs, employee–organization, employee–supervisor, and employee–coworker communications could be conducted online. Previous research determined that the benefits of social capital are not only manifested in incremental improvements but also in more dramatic advances in business processes, which is closely related to job satisfaction [52]. Leftheriotis et al.’s (2014) organization research explained that positive co-worker relationships and social capital positively influenced job satisfaction [30]. Furthermore, Requena (2003) found a positive relationship between social capital and job satisfaction [53]. SNSs help chefs to have social interaction or communication with others, which impacts their social capital with other and assists in the development of their job satisfaction. Thus, this study argues that social capital positively influences job satisfaction.

**Hypothesis** **4 (H4).**
*Social capital bonding is positively associated with job satisfaction.*


**Hypothesis** **5 (H5).**
*Social capital bridging is positively associated with job satisfaction.*


### 2.6. Moderating Effect of Customer Orientation

Customer orientation refers to the degree to which a service worker practices the marketing concept by trying to help customers make purchase decisions or their experience that satisfies their needs [21]. In the marketing research, a previous study found that customer-oriented workers will engage in behaviors directed at value creation and relationship development with customers [54]. Guenzi et al.’s (2011) study demonstrated that customer orientation is an important predictor of job performance and job satisfaction [22]. In this perspective, the customer orientation of service workers is a critical factor in positively impacting the finance of the firm.

Merlo et al. (2006) and XueMing et al. (2004) demonstrated a positive relationship among social capital, customer relationship, and customer orientation [52,55]. Thus, chefs who show high degrees of customer orientation also positively influence their job satisfaction more than the low customer orientation group. Another study determined that customer orientation influences service worker job satisfaction [56]. Thus, this study hypothesized that customer orientation has a moderation effect on the path. The hypothesis is as follows.

**Hypothesis** **6 (H6).**
*Customer Orientation has a moderation effect on the paths.*


Our proposed research model, based on previous studies, is presented in Figure 1. This conceptual model includes the social support from the chef SNS community (bonding and bridging), social presence, job satisfaction, and customer orientation. Thus, a total of six research hypotheses related to the relationships among these variables were formulated (H1–H6).

## 3. Material and Methods

### 3.1. Study Design and Participants

The objective of this study was to examine how bonding social capital and bridging social capital enhance social presence, which impacts job satisfaction. In addition, the moderation effect of customer orientation was also investigated. The data process used and choice of sample respondents are outlined as follows. According to a 2019 report, there are 460 hotels located in Seoul, South Korea. Only 24 hotels rank as 5-star. This study also considered whether they are Michelin-starred restaurants. Considering the objective of this study, we chose chefs that work in Michelin-starred restaurants with sales in the top five. The reasons we chose these Michelin-starred restaurants are as follows: First, social capital refers to the social relationship between employee and organization, which may impact a firm’s financial performance. For larger hotels, the organizational structure has a strong influence. Second, Michelin-starred restaurants concentrate on offering consumers hedonic value, enjoyment experience, and emotion, which require service workers to have a high customer orientation. Thus, this study chose chefs working in Michelin-starred restaurants as our sample. With the assistance of a classmate who worked at S-Hotel in Seoul, researchers met with the cooking department officer to ask permission to conduct the survey. This study used a clear question to confirm that all survey participants used chef SNS communities to develop relationships with other chefs and seek cooking recipes, such as Naver chef café or Facebook chef pages. Researchers then invited these chefs to participate in the survey; 180 hotel chefs were invited to participate in this study from 22 July to 5 August 2017. Finally, we received 153 responses from chefs who were SNS users and employed SNS to have social interactions with others. A total 130 responses that did not contain missing values were used for statistical analysis. Demographic information consisted of gender, age, work experience as a 5-star hotel chef, education, and SNS usage behavior. The specific profiles of the respondents were presented as follows. Respondents consisted of 100 male (73%) and 30 females (27%). A total of 130 (19.7%) were aged 10–19, while 28 (20.4%) were aged 20–29, 26 (19.0%) aged 30–39, 24 (17.5%) aged 40–49, and 32 (23.4%) aged 50–59. For the question about their career in 5-star hotel, most (40) had careers of 1–5 years (29.2%), 31 had careers of 6–10 years (22.6%), 29 over 16 years (21.2%), 20 less than 1 year (14.6%), and 17 11–15 years (12.4%). Most, i.e., 69, graduated from 2-year college (50.4%), 57 (41.6%) from 4-year college, and 11 (8%) graduated from high school. SNS usage results are as follows: 56 (40.9%) 1–3 years, 41 (29.9%) less than 1 year, 26 (19.0%) 3–5 years, 7 (5.1%) 5–7 years, and 7 (5.1%) more than 7 years.

### 3.2. Measures

A self-administered questionnaire was developed for the survey, which included bonding social capital, bridging social capital, social presence, job satisfaction, and demographic information. All measurement items are listed in Table 1. Bonding social capital refers to strong ties with others, who promote empowerment and emotionally close relationships within the group. Bonding social capital measurement was adopted from Adler and Kwon (2002) and Lee (2013) [57,58]. Bridging social capital was defined as the connections between people of different hierarchies or ages, and is usually associated with weak ties, which are loose connections between individuals rather than emotional support. Bridging social capital consisted of four items, developed from Ellison et al. (2007) and Lee (2013) [46,57]. Social presence is a sub-dimension construct which includes mutual perception, emotional bondage, and collective space, and was defined as interactions in a virtual environment that is the same as in the physical environment. Social presence was measured with three items, adopted from Han et al. (2016) and Gefen and Straub (2003) [59,60]. Job satisfaction was defined as the individual’s degree of satisfaction in their job, which was measured with four items from Kathrin et al. (2013) [61]. Customer orientation refers to the degree to which service workers try to help customers make purchase decisions or create an experience that will satisfy their needs [21]. The measurements were developed from Liaw and Chuang (2009) and Donavan et al. (2004). The five items were “I believe that providing timely, efficient service to customers is a major function of my job.”; “I enjoy nurturing my service customers.”; “I take pleasure in making every customer feel like he/she is the only customer.”; “Every customer’s problem is important to me.”; and “I thrive on giving individual attention to each customer.”

All measurements were first developed in English and translated into Korean by this study’s researcher, who is fluent in both English and Korean. Then, the Korean version of the survey questionnaire was back-translated into English by an English professor who is a native speaker of Korean to ensure questionnaire translation equivalence. To verify the questionnaire validity and fitness, this study conducted pre-tests from 20 June to 10 July 10 2017. The pre-test results indicated that there were no problems concerning the questionnaire validity. All of the items were measured on a five-point Likert scale ranging from one (strongly disagree) to five (strongly agree). The questionnaire is presented in Table 1.

### 3.3. Statistical Analysis

The purpose of this study was to examine the factors involved in hotel chef job satisfaction. SPSS was used for analyzing the data reliability. Cronbach’s alpha and AMOS were applied for measurement instrument validity. AMOS was also used for structural equation modelling, which assists in testing a multifaceted model concurrently. Therefore, this study used SPSS 18.0 (IBM Corp., Armonk, NY, USA) and AMOS 25.0 (IBM Corp., Armonk, NY, USA) software to conduct statistical analysis. SPSS 18.0 was used for demographic information analysis and exploratory factor analysis. Analyses of confirmatory factor, path, mediation effect, and moderation effect were conducted using AMOS 25.0.

## 4. Results

### 4.1. Measurement Model

Both an exploratory factor analysis (EFA) and confirmatory factor analysis (CFA) were used in this study to analyze the measurement model. First, this study conducted EFA using principal component analysis varimax rotation. All of the items were analyzed for four factors. The significant Keiser–Meyer–Olkin value was less than 0.001 (*p* < 0.001). The factor loadings for measurements on the intended constructs were higher than 0.5. Furthermore, the value of Cronbach’s alpha was higher than 0.8, which indicated that the scale had good internal consistency.

Then, this study conducted CFA to test the measurement model. Contents validity, convergent validity, and discriminant validity were used to indicate the measurement. The measurement model in general fit the data adequately: X^2^ = 267.845, *p* < 0.001, goodness of fit index (GFI) = 0.898, root mean square error of approximation (RMSEA) = 0.08, RMR = 0.047, normed fit index (NFI) = 0.913, and comparative fit index (CFI) = 0.941 [62]. See Table 2. Convergent validity and discriminant validity are important goals of CFA to determine construct validity in structure modeling. Convergent validity is assessed by the value of factor loading. Previous studies suggested that values higher than 0.5 are desirable. The average variance extracted (AVE) measures verified that the indicator is suitable, with a value higher than 0.5. Table 2 shows that all variables were retained as they satisfied the acceptability guidelines. Composite reliability (CR) for each latent construct was applied to measure the internal consistency of indicators. According to Hair et al. (2010), a value of CR higher than 0.7 indicated strong composite reliability [62]. The CR value of this study exceeded 0.7, indicating good reliability. See Table 1. 

Discriminant validity determines whether one construct is indeed different from other constructs, and that there is no overlap of measuring indicators between one construct and another. One way to verify discriminant validity is to examine the correlations of the different variables. A more stringent method was used to verify discriminant validity, i.e., by comparing the average AVE values of any two variables with the squared correlation estimated between these variables. The average AVE value of two variables should exceed the squared correlation estimate. The results of discriminant validity are presented in Table 2.

### 4.2. Structural Model and Discussion

The structural equation modeling show that this research model had an acceptable fitness with X^2^ = 267.845, GFI = 0.898, RMSEA = 0.08, RMR = 0.047, NFI = 0.913, and CFI = 0.942. See Table 3. Testing hypotheses 1 and 2 showed that social capital bridging significantly impacted social presence (β = 0.693, *p* < 0.001). On the contrary, social capital bonding influenced social presence, which was not significant (β = −0.178, *p* > 0.05). Thus, H1 was not supported, but H2 was supported. Social presence significantly impacted job satisfaction (β = 0.129, *p* < 0.01). In addition, both social capital bonding (β = 0.318, *p* < 0.01) and bridging (β = 0.240, *p* < 0.01) significantly influenced job satisfaction, which support H3–H5. See Table 3, Figure 2.

### 4.3. Moderation Effect

This study also investigated the moderation effect of customer orientation. To examine the moderation effect, this study divided customer orientation into a high and low groups (high customer orientation group vs. low customer orientation group). This study compared unconstrained and constrained models. The unconstrained model’s factor structure was specified as varying across groups, while the factor structure was constrained to be the same across groups. If the X^2^ fit difference between these models is insignificant, the factor structure is invariant across groups. As presented in Table 4, the results of the likelihood maximum estimation method of the AMOS software indicates that customer orientation showed a significant difference in the relationship between social capital bonding and social presence (the variation of X^20.05^ = 4.361), and the relationship between social capital bonding and job satisfaction (the variation of X^20.05^ = 4.802). Thus, H6 is partially supported. Table 5 and Figure 3 present the specific results of the moderation effect.

### 4.4. Mediation Effect

The Sobel test and Hayes (2013) argued bootstrapping was used to test the mediation effect [63]. Social presence was found to have a significant mediation effect between social capital and job satisfaction (z = 2.221, *p* < 0.05). The result of the bootstrapping coefficient was −0.041, which is in the range of −0.097 to −0.009. Thus, the full mediation effect was verified. Social presence was demonstrated to have a significant mediation effect between social capital bridging and job satisfaction. The coefficient value was −0.077, which is in the −0.015 to −0.020 range. Thus, the full mediation effect was verified. The mediation effect results are presented in Table 6.

## 5. Conclusions

This study aimed to demonstrate how the chef social community helps in maintaining chef’s social capital, which can maintain hotel restaurant sustainability in terms of economic development. This study explored the social capital of the chef SNS community regarding social presence and job satisfaction. The chef SNS community was found to be helpful in enabling chefs to develop relationships and foster recipe knowledge exchange. Following previous research, this study developed a model to investigate the relationship among these variables. This study invited 130 chef SNS community respondents to participate in our survey for hypotheses testing. This study used SPSS and AMOS to conduct statistical analysis. Five hypotheses were determined to have a significant relationship. To be specific, the results of this study indicated that bridging social capital significantly positively influences social presence and job satisfaction, while the influence of bonding social capital did not significantly affect social presence. The results also determined that bonding and bridging social capital significantly influence job satisfaction. Following these results, only H2 was not significant. Other hypotheses presented significant results. In addition, this study found that social presence significantly impacts job satisfaction. Thus, H5 was meaningful. From the analysis of the moderation effect of customer orientation, the results also predicted that customer orientation produces a notable moderation effect. H6 was thus confirmed. Based on these results, this study could offer theoretical implications for future studies and has managerial implications for hotel managers to enhance chef job satisfaction.

## 6. Implications

### 6.1. Theoretical Implications

First, previous studies paid less attention to the benefits of chef SNS community participants. This study applied social capital theory and social presence theory to determine whether the effect of chef SNS participants positively impacts their job performance and job satisfaction. In addition, this study extended the original model by adding chefs’ customer orientation to test the customer orientation moderation effect. Chef SNS community participants help chefs to develop social relationships and exchange cooking knowledge. The results indicated that bridging social capital was significantly associated with social presence and chef job satisfaction. Previous studies also determined that bridging social capital is positively related to social presence and chef job satisfaction [30,35]. However, the bonding social capital did not significantly influence social presence. Previous studies argued that bonding social capital also impacts social presence. This study explored the different results, which can be explained by the fact that chefs’ work requires strong relationships. When a customer order is printed, every chef has to accurately fulfil their role to ensure the menu is delivered well. Weak relationships do not therefore significantly influence chefs’ social presence [35]. Thus, this result is also meaningful. This study also explored the influence of social presence on job satisfaction. The results showed that social presence positively impacts chef job satisfaction. These results are consistent with those of prior studies, because they illustrate that social presence enhances chefs’ sense of being together and helps improve chefs’ job satisfaction [46].

Second, this study added a moderator, customer orientation, to extend the model. Chefs are required to satisfy customer needs and provide customers with an enjoyable experience. High customer orientation chefs continue to develop their cooking skill to offer a better experience for the customer more than low customer orientation chefs. This development is represented as active communications with other chefs to develop social capital in the SNS community. The results showed a significant difference in the path, which is consistent with previous research [56].

This study confirmed the role of the SNS community on the experience of hotel chefs. Previous studies were less focused on the role of the SNS community for hotel chefs. This study confirmed that the chef SNS community plays an important role in assisting them in developing social relationships to enhance their job satisfaction and performance. Thus, this study offered insight for further study. The current study confirmed social capital positively impacts chefs’ job satisfaction. Further studies could consider social capital as a factor to expand other research models. In addition, further research could consider the effect of social capital in other service industries. The current study also confirmed the moderation effect of customer orientation. Customer orientation is a critical factor in determining job satisfaction in the service industry. Further studies could be conducted based on these findings.

### 6.2. Managerial Implications

From a practical standpoint, the results of this study could provide a strategy for human resources management. The food and beverage industry has critical factors which have positive economic effects in hotel management. For sustainable economic development, this study could offer chef SNS community guidelines for managers, which could affect chef job satisfaction and reduce chef turnover. First, it is important to assist chefs to join the chef SNS community. After joining the SNS community, chefs can more easily develop social relationships with other chefs, whether strong relationships or weak relationships. According to the results of this study, both bridging social relationships and bonding relationships are positively correlated to chefs’ job performance and job satisfaction. Thus, helping chefs to participate in the SNS community is essential. For instance, chef managers could build a community and invite their team to join this community. Chefs would then manage their team relationship more effectively, which would greatly improve teamwork. Thus, it could help the formation of strong relationships in the kitchen. Furthermore, hotel managers could gather to build a large community. In this community, chefs could exchange recipes that were well-received by customers, thus assisting chefs to exchange their recipe knowledge. This would have the result of enhancing chefs’ cooking skills, job performance, and job satisfaction. This may be a weak relationship, in which the SNS community could enhance chefs’ satisfaction to support sustainable economic development.

Second, improving chef customer orientation is important. A chef’s job is to offer service to customers to provide them with an experience and satisfy their needs. Thus, it is important to improve chefs’ customer orientation. High customer orientation chefs wish to enhance the experience of their customers and improve the taste of food. This orientation will positively impact chefs to communicate with other chefs and learn cooking skills, thus improving their job performance. Therefore, chef managers should continue to train their chefs to maintain their service orientation.

Third, the results of this study could provide managerial implications for service employees. Chefs, as a type of service employee, have the same characteristics as other service employees. Human resources managers should therefore help service workers to build the SNS community and also assist service employees to build and enhance relationships with other service workers. Through this online community, they could develop strong relationships with others and enhance their job satisfaction.

### 6.3. Limitations

Based on a survey, this study found that the SNS community social capital could improve chefs’ social presence and job satisfaction. However, this study has certain limitations. To address these, we recommend the following should be considered in future research on the chef SNS community behavior. First, this study’s findings cannot be generalized to a large population of chefs because it utilized data from only 5-star hotels in Seoul. Moreover, this study’s participants were not associated with demographic factors such as education and work career. Future studies should collect more sample data for statistical analysis and analyze the effect of demographic factors on job satisfaction. Second, social presence consists of several dimensions. This study used a single dimension to define social presence. Future studies should apply multiple dimensions to expand the social presence research model [35]. Finally, this study determined that SNSs play a positive role in the chef’s job. Future studies should consider more SNS variables to test the relationship.

## Figures and Tables

**Figure 1 ijerph-17-07129-f001:**
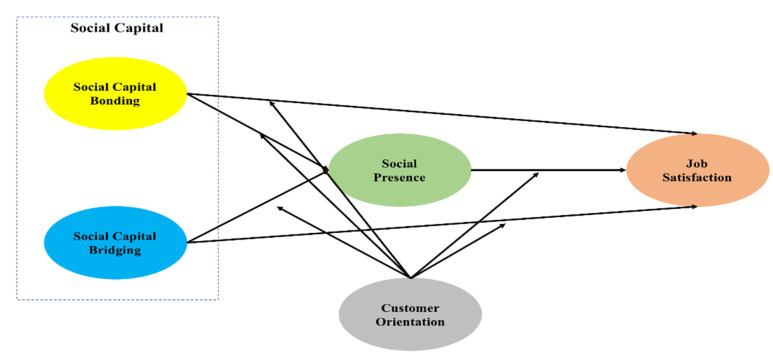
Research model.

**Figure 2 ijerph-17-07129-f002:**
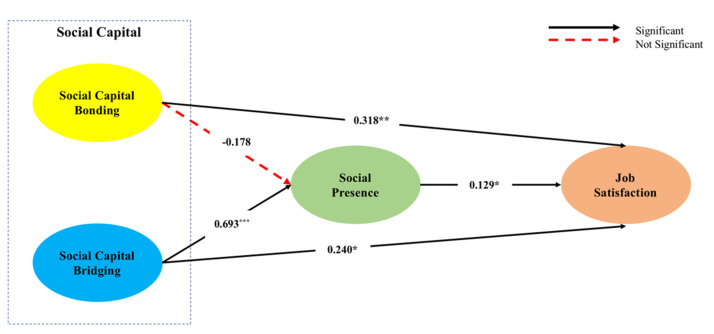
Results of hypotheses. Model Fit: X^2^ = 267.845, GFI = 0.898, RMSEA = 0.08, RMR = 0.047, NFI = 0.913, CFI = 0.942. *** *p* < 0.001; ** *p* < 0.01; * *p* < 0.05.

**Figure 3 ijerph-17-07129-f003:**
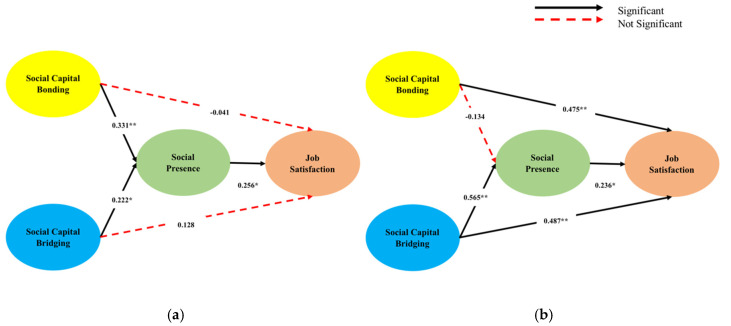
Results of moderation effect. * *p* < 0.05; ** *p* < 0.01. (**a**) Low group *N* = 64. (**b**) High group *N* = 66.

**Table 1 ijerph-17-07129-t001:** Confirmation factor analysis.

Construct	Measurement	Factor Loading	Cronbach’s α	CR	AVE
Social Capital Bridging (SCBR) [57,58]	SCBR1: I use this SNS for cooking information because interacting with people whom I do not know in real life makes me interested in what people unlike me are thinking	0.778	0.913	0.899	0.691
	SCBR2: I am willing to spend time on this SNS supporting the cooking information of people whom I do not know in real life	0.873			
	SCBR3: I use this SNS for cooking information because interacting with people whom I do not know in real life makes me want to try new things	0.866			
	SCBR4: I use this SNS for cooking information because interacting with people whom I do not know in real life makes me feel like part of a larger community	0.885			
Social Capital Bonding (SCBO) [46,57]	SCBO1: I use this SNS for cooking information because there is someone, whom I do not know in real life, who I can turn to for advice about making very important decisions	0.818	0.910	0.903	0.652
	SCBO2: use this SNS for cooking information because the people whom I do not know in real life but with whom I interact with on this SNS would share their last dollar with me	0.841			
	SCBO3: I use this SNS for cooking information because the people whom I do not know in real life but with whom I interact with on this SNS would put their reputations on the line for me	0.774			
	SCBO4: I use this SNS for cooking information because the people whom I do not know in real life but with whom I interact with on this SNS would help me fight an injustice	0.819			
	SCBO5: I use this SNS for cooking information because there are several people on this SNS whom I do not know in real life but who I trust to help solve my problems	0.840			
Social Presence (SP) [59,60]	SP1: There is a sense of human contact in SNS	0.869	0.867	0.854	0.661
	SP2: There is a sense of personalness in SNS	0.801			
	SP3: There is a sense of sociability in SNS	0.813			
Job Satisfaction (JS) [61]	JS1: How satisfied are you with your overall job?	0.638	0.827	0.863	0.616
	JS2: How satisfied are you with your immediate supervisor?	0.916			
	JS3: How satisfied are you with your organization’s policies?	0.749			
	JS4: How satisfied are you with the support provided by your supervisor?	0.680			
	Model fit: X^2^ = 267.845, GFI = 0.898, RMSEA = 0.08, RMR = 0.047, NFI = 0.913, and CFI = 0.942.				

CR = Composite reliability; AVE = Average variance extracted; GFI = Goodness of fit index; RMSEA = Root mean square error of approximation; RMR = Root mean square residual; NFI = Normed fit index; and CFI = Comparative fit index.

**Table 2 ijerph-17-07129-t002:** Discriminant validity.

Construct	Descriptive Statistics		Discriminant Validity			
	M	SD	SCBO	SCBR	SP	JS
SCBO	2.84	0.98	1			
SCBR	2.99	0.90	0.771 ***	1		
SP	3.12	0.94	0.383 ***	0.491 ***	1	
JS	3.34	0.73	0.208 ***	0.123 *	0.131 *	1

*** *p* < 0.001; * *p* < 0.05, SCBO = Social capital bonding; SCBR = Social capital bridging; SP= Social presence; JS = Job satisfaction.

**Table 3 ijerph-17-07129-t003:** Results of hypothesis pathway.

Hypothesis	Coefficient	Std. Error	T-Value	Support
H1: Bonding-Social Presence	−0.178	0.154	−1.152	No
H2: Bridging-Social Presence	0.693	0.137	5.054	Yes
H3: Social Presence-Job Satisfaction	0.129	0.057	2.276	Yes
H4: Bonding-Job Satisfaction	0.318	0.116	2.739	Yes
H5: Bridging-Job Satisfaction	0.240	0.109	2.205	Yes
Model Fitness: X^2^ = 267.845, GFI = 0.898, RMSEA = 0.08, RMR = 0.047, NFI = 0.913, CFI = 0.942.				

GFI = Goodness of fit index; RMSEA = Root mean square error of approximation; RMR = Root mean square residual; NFI = Normed fit index; and CFI = Comparative fit index.

**Table 4 ijerph-17-07129-t004:** Results of moderation effect.

Default Model	X^2^	DF	ΔX^2^ (df = 1)
Free Model	410.260	196	
H1: Bonding-Social Presence	414.621	197	4.361
H2: Bridging-Social Presence	412.858	197	2.598
H3: Social Presence-Job Satisfaction	410.271	197	0.011
H4: Bonding-Job Satisfaction	415.063	197	4.803
H5: Bridging-Job Satisfaction	412.781	197	2.521

**Table 5 ijerph-17-07129-t005:** Results of moderation effect.

Path	High Customer Orientation			Low Customer Orientation		
	Coefficient	T	*p*	Coefficient	T	*p*
Bonding-Social Presence	−0.134	−0.731	0.465	0.331	2.882 **	0.004
Bridging-Social Presence	0.565	3.028 **	0.002	0.222	1.969 *	0.049
Social Presence-Job Satisfaction	0.236	1.982 *	0.05	0.256	2.199 *	0.028

** *p* < 0.01; * *p* < 0.05.

**Table 6 ijerph-17-07129-t006:** Mediation effect.

Dependent Variable	Independent Variable	Mediating Variable	Indirect Effect	Sobel Test	Confidence Interval 95%	
					LLCI	ULCI
Job Satisfaction	Social capital bonding	Social Presence	−0.041	2.211 *	−0.097	−0.009
	Social capital bridging		−0.077	2.386 *	−0.155	−0.020

* *p* < 0.05; LLCI = Lower limit confidence interval; ULCI = Upper limit confidence interval.

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
