# Peer review of "What Makes Hotel Chefs in Korea Interact with SNS Community at Work? Modeling the Interplay between Social Capital and Job Satisfaction by the Level of Customer Orientation"

_ijerph, 2020, doi:10.3390/ijerph17197129_

Round 1
Reviewer 1 Report
Please refer to the appended file.

Reviewer 2 Report
Overall, this study has displayed solid usage of quantitative methods. The structure is mostly organized and clear. There are some areas where the explanations and language can be strengthened. Meanwhile, although the data are novel, the generalizability of findings may be limited as the authors have already noted in the manuscript. My detailed comments and questions can be found below.
- “As the previous study argued, as customer orders stream out of the printer on the pass table the head chef calls them out to the sous chefs who, together with commis chefs, immediately begin preparing the meals”. Consider rewording.
- “The chef SNS community is also regarded as [an] important way to develop relationships with other chefs and facilitate recipe knowledge exchange”. There are a few other sentences in the manuscript that need editing.
- “Following these previous studies, social capital is a critical factor in influencing employee social presence and job satisfaction. Chef SNS community social capital could assist chefs to reduce the stress they face in the work environment, which will positively impact their social presence and job satisfaction”. This sounds like a stretch. Any previous research to support this claim? Or did the authors try to state a hypothesis here? If it’s the latter, the authors may want to clearly label this sentence as a hypothesis.
- “Previous studies have demonstrated that service worker consumer orientation differently impacted their job satisfaction [22]. However, these studies have paid little attention to investigate the difference in customer orientation effect.” I think the authors were contradicting themselves here. Did they try to say that: these studies have paid little attention to investigate the difference in customer orientation moderation effect instead?
- “Following the Ozatok Murat et al., (2015) research, they employed social capital theory to understand the social presence of students in online learning environments and illustrated that social presence related more to communication between weak ties rather than within strong-tied subsets of participants [35]”. Then in H1, the authors stated that bonding social capital is positively associated with social presence, which is basically the same with H2’s hypothesized relationship surrounding the bridging social capital. Is there some other way to distinguish these two kinds of social capital and their impacts on the outcome variable in the hypotheses? Should these two hypotheses be worded differently to better suggest differences in effect size and/or direction?
- “Social presence theory argues that a quality of the medium itself and assert media varies in the degree of social presence”. Consider rewording.
- The outcome variable in H1 and H2 is social presence. Doesn’t it make more sense to introduce the readers to the definition of social presence before H1 and H2 instead of afterwards in section 2.4?
- The wording of H6 can be improved: effects “on” not “in” and “paths” instead of “path” as the following figure shows that the moderating effects may be found on more than one path.
- “to participant our survey” should be “participate”.
- When looking at Table 1, the items for social bonding capital read like those for bridging which emphasize distant, more generic relationships, while those for bridging social capital seem to be a better fit for bonding which suggest closeness and trust. Can the authors double check on the labeling of those items?
- How is customer orientation measured in this study? The measurement of this important variable was neither discussed in text nor in table 1.
- “This chose SPSS and AMOS 53 to finish statistical analysis.” Consider editing. There are a few other sentences in the conclusions section that need to be edited.
- The authors may also want to discuss how their research on SNS might be relevant to contemporary times, i.e., the pandemic when SNS has become an increasingly important tool for social capital, social presence, and perhaps job satisfaction. This might help the authors better sell their study to a broader audience, especially given the fact that the sample is geographically and professionally limited.
- With regard to SNS, what kind of platforms did the survey participant use, Facebook or something else? And how frequently did they use SNS? Is SNS usage solely measured by the length of usage as described in the methods section?
Reviewer 3 Report
The article is well written and prepared. The topic is topical and interesting. The structure of the article is clear. The methodology used is relevant to the topic. The authors presented the results of research aimed at demonstrating the impact of the social capital of restaurant chefs on their social presence and job satisfaction. Research questions and hypotheses were correctly formulated. The survey was conducted among chefs working in restaurants awarded with a Michelin star in Seoul, Korea. The importance of ANN as an important means of creating relationships and sharing knowledge was emphasized. Exploratory (EFA) and confirmatory (CFA) factor analysis were used. The discussion section is well structured. The authors indicated both theoretical and practical implications. The conclusions may be of interest to the reader. The bibliography is current and contains many items.
Round 2
Reviewer 2 Report
The authors have sufficiently addressed my comments and questions.